# Dissecting Genetic Mechanisms of Differential Locomotion, Depression, and Allodynia after Spinal Cord Injury in Three Mouse Strains

**DOI:** 10.3390/cells13090759

**Published:** 2024-04-29

**Authors:** Wendy W. Yang, Jessica J. Matyas, Yun Li, Hangnoh Lee, Zhuofan Lei, Cynthia L. Renn, Alan I. Faden, Susan G. Dorsey, Junfang Wu

**Affiliations:** 1Department of Anesthesiology and Shock, Trauma and Anesthesiology Research Center (STAR), University of Maryland School of Medicine, Baltimore, MD 21201, USA; wendyyang@usf.edu (W.W.Y.); jmatyas@rochesteru.edu (J.J.M.); yun.li@som.umaryland.edu (Y.L.); zlei@som.umaryland.edu (Z.L.); afaden@som.umaryland.edu (A.I.F.); 2Department of Medicine, University of Maryland School of Medicine, Baltimore, MD 21201, USA; hangnoh.lee@som.umaryland.edu; 3Department of Pain and Translational Symptom Science, University of Maryland School of Nursing, Baltimore, MD 21201, USA; renn@umaryland.edu (C.L.R.); sdorsey@umaryland.edu (S.G.D.)

**Keywords:** spinal cord injury, mouse strains, neuropsychiatric behaviors, genetics, RNA sequencing, genomics

## Abstract

Strain differences have been reported for motor behaviors, and only a subset of spinal cord injury (SCI) patients develop neuropathic pain, implicating genetic or genomic contribution to this condition. Here, we evaluated neuropsychiatric behaviors in A/J, BALB/c, and C57BL/6 male mice and tested genetic or genomic alterations following SCI. A/J and BALB/c naive mice showed significantly less locomotor activity and greater anxiety-like behavior than C57BL/6 mice. Although SCI elicited locomotor dysfunction, C57BL/6 and A/J mice showed the best and the worst post-traumatic recovery, respectively. Mild (m)-SCI mice showed deficits in gait dynamics. All moderate/severe SCI mice exhibited similar degrees of anxiety/depression. mSCI in BALB/c and A/J mice resulted in depression, whereas C57BL/6 mice did not exhibit depression. mSCI mice had significantly lower mechanical thresholds than their controls, indicating high cutaneous hypersensitivity. C57BL/6, but not A/J and BLAB/c mice, showed significantly lower heat thresholds than their controls. C57BL/6 mice exhibited spontaneous pain. RNAseq showed that genes in immune responses and wound healing were upregulated, although A/J mice showed the largest increase. The cell cycle and the truncated isoform of trkB genes were robustly elevated in SCI mice. Thus, different genomics are associated with post-traumatic recovery, underscoring the likely importance of genetic factors in SCI.

## 1. Introduction

Spinal cord injury (SCI) results not only in debilitating motor, sensory, and cognitive deficits, but also in chronic neuropsychiatric disorders that contribute to life-long suffering and decreased quality of life for these patients. To date, no neuroprotective or neurorestorative therapy has clearly improved, clinically, long-term recovery. This may reflect, in part, an incomplete understanding of injury mechanisms, including the role of genetic differences. The latter can significantly impact the response to SCI across mouse strains. The Basso Mouse Scale for locomotion (BMS) following SCI showed differences across five mouse strains commonly used in experimental research [1], strongly suggesting that inherent genetic factors may play a role in the behavioral profiles. However, no studies to date have characterized overall neurological functions across several strains of mice after SCI. This is important as SCI-associated depression and neuropathic pain can adversely affect rehabilitation and motor recovery.

The severe, persistent central neuropathic pain that occurs in SCI patients is highly resistant to conventional pain treatment. Only a subset of SCI patients develop neuropathic pain and large interindividual differences have been observed in SCI-induced neuropathic pain [2], implicating the genetic or genomic contribution to this condition. It is likely that susceptibility to neuropathic pain development and/or severity may be at least partially inherited. Thus, there is a critical need for the development of genetic approaches to understand pain-generating and pain-maintaining mechanisms related to SCI-PAIN. Although strain differences in pain-related anatomy, electrophysiology, and neurochemistry have been observed [3,4,5], to our knowledge, no studies to date have characterized SCI-PAIN across several strains of mice commonly used in SCI research.

We previously characterized the genomics of rodent SCI extensively, which resulted in studies on the role of cell cycle activation and delayed neuroinflammation in the pathophysiology of SCI [6,7,8]. More recently, transcriptomic analysis identified cell cycle gene regulation as a possible downstream target of trkB.T1 signaling in mouse models of neuropathic pain, which is a truncated isoform of the tropomyosin related kinase B (trkB) receptor for brain-derived neurotrophic factor (BDNF) [9,10]. Thus, genomic analyses associated with SCI-mediated neurological dysfunction may help to clarify potential therapeutic targets. However, little attention has been paid to comparative genetics and genomics between mouse strains following SC

The purpose of the present study was to characterize behavioral changes in locomotion, allodynia, and depression across several strains of mice (A/J, BALB/c, and C57BL/6) commonly used in SCI research and to examine differential gene expression across these three mouse strains. We show, across different mouse strains subjected to similar impact injuries, significant differences in locomotor recovery, anxiety/depression-like behaviors, and tactile allodynia. RNAseq analyses show differential post-traumatic gene expression and pathway regulation in these strains, including dysregulated cell cycle and trkB.T1 genes that are known to be involved in SCI-pathophysiology.

## 2. Materials and Methods

### 2.1. Mice and Spinal Cord Injury

Young adult male mice of the A/J, BALB/c, and C57BL/6 inbred strains were purchased from Jackson Labs at 9 to 12 weeks of age. All animals were housed in the University of Maryland School of Medicine animal facility in a controlled environment with a 12 h (6 a.m. to 6 p.m.) light/dark cycle and ad libitum access to food and water at 4 to 5 mice per cage. Following isoflurane anesthesia, mice were subjected to a laminectomy at the T9 to T11 region, and the spinal column was placed onto bilateral steel clamps for stabilization. The mice received T10 spinal cord contusions with 35 (mild injury, m) or 60 (moderate/severe injury, mo/se) kDyne force using the Infinite Horizon Spinal Cord Impactor (Precision Systems and Instrumentation) [11,12]. Manual bladder expression was performed on SCI mice a minimum of three times per day in the first two weeks of injury until reflex bladder emptying was re-established. Sham mice underwent the same procedure except for contusion. All surgical procedures and behavioral experiments performed were under protocols approved by the University of Maryland School of Medicine Institutional Animal Care and Use Committee.

### 2.2. Behavioral Assessment

Open field (OF): This test was used to assess spontaneous motor activity and anxiety-like behavior under dim light [11,12]. At the start of the test, each mouse was individually placed in a corner facing the wall of the open-field arena (40 cm × 40 cm). The mice were allowed to explore the chamber freely for a total duration of 5 min (min). On Any-maze, the open field apparatus was divided into a grid of 10 cm × 10 cm squares. The outer 10 cm zone was defined as the outer zone, whereas the center 20 × 20 cm^2^ region was defined as the inner zone. Parameters such as total distance travelled, average and max speed, and time spent in inside and outside zones were recorded with the Any-maze tracking program (Stoelting Co., Wood Dale, IL, USA).

Basso mouse scale (BMS) for locomotion: For assessment of hindlimb locomotion recovery, mice were observed for at least 4 min by two trained observers after being placed in a flat, enclosed box (62 cm × 42 cm). Both observers were blinded to the genetic background of the mice. Mice were tested for BMS scores on day 1 and 3, followed by weekly assessments for up to 8 weeks post injury. The BMS was used to score hindlimb locomotion on a scale of 0–9 based on joint movement, weight support, plantar stepping, and coordination of the hind limbs, with 0 being complete paralysis and 9 being normal function [1].

CatWalk XT automated gait analysis: Gait analysis with CatWalk XT (Noldus, http://www.noldus.com/, accessed on 26 April 2024; RRID: SCR_004074) was performed in a darkened room with the protocol as mentioned in previous publications [10,13]. The CatWalk apparatus features a green illuminated walkway and records multiple parameters with a fast-action camera placed directly under the glass walkway. Parameters recorded include regularity index, print position, stride length, weight distribution, and gait postures. A single researcher blinded to the group information was tasked with animal handling and data analysis. At least three valid runs or complete walkway crossings were obtained for each mouse. Runs with mid-way stops, turns, or scaling of sidewalls were considered invalid and excluded from analysis.

Tail suspension (TS) test: This test was used to assess depressive-like behavior on week 8 as described previously [11,12]. Briefly, each mouse was suspended at a height of 28 cm with 3 M adhesive tape placed approximately ~1 cm from its tail tip. A timer of 5 min was set for the test period, while the duration of time spent immobile (passive hanging, motionlessness) was recorded with a stopwatch.

Forced swim (FS) test: For assessment of depressive-like behavior, we performed FS test as previously described [11,12]. Mice were placed in a transparent glass cylinder (25 cm high and 20 cm diameter) filled with water (22 cm in depth) at room temperature (22–23 °C). The duration of time spent immobile, which is characterized as lack of vigorous activity and movement only to maintain its head above water, was recorded with a stopwatch throughout the 6 min test period.

Von Frey filament test: For determination of mechanical allodynia development, the von Frey filament test was used to detect hind paws withdrawal from a mechanical stimulus according to a method published previously [9,14]. Prior to testing, each mouse was placed in individual plexiglass cubicles on a wire mesh platform and underwent an acclimation period of 1 h. The von Frey filaments (MyNeuroLab, St. Louis, MO, USA), which have an incremental stiffness ranging from 0.04 to 2.0 g, were applied serially to the plantar surface of each hind paw in ascending order. A total of five von Frey filament applications were made perpendicularly to the hind paw in each trial, each lasting 2–3 s and with a 5 s interval between each application. A positive response is defined as brisk withdrawal of the paw (at least 3 times out of 5 applications) upon application of the filament. Whereas a negative response was defined as no paw withdrawal, and the filament was changed to the next one in ascending stiffness. Furthermore, the application of a filament was repeated when paw response was ambiguous. The threshold was defined as the filament with the lowest bending force that elicited at least 3 positive responses out of 5 trials.

Hot plate test: Thermal allodynia of the hindpaws was tested with an Incremental Hot/Cold Plate Analgesia Meter (PE34, IITC Life Science, Woodland Hills, CA, USA) as previously published [9,14]. In brief, mice were placed within a transparent plexiglass box (22 cm length, 12 cm width, and 20 cm height) on the contact probe of a computerized thermal stimulator. The plate temperature was set to increase from 30 to 50 °C with the incremental ramp rate set at 10 °C per minute. Testing was stopped when the mouse licked either one of its hind paws, and the threshold temperature was recorded. The test was conducted twice with an interval of 3 h between the first and second trials.

### 2.3. Tissue RNAseq Analysis

At the study endpoint, anesthetized mice were transcardially perfused with ice-cold normal saline. Mouse spinal cord tissue (~0.5 cm) centered on the injury site was dissected out and fast-frozen in dry ice. Total RNA was extracted using an RNeasy mini kit (Cat# 74104, Qiagen, Hilden, Germany) with the concentration measured and sent to the Institute for Genome Sciences (IGS) at the University of Maryland School of Medicine for RNA sequencing [9,10].

Library construction and sequencing: Sequencing libraries were prepared by the IGS with the Illumina mRNA-Seq Sample Prep Kit (Illumina, San Diego, CA, USA. #RS-122-2101), followed by sequencing on an Illumina HiSeq4000 Sequencer that used the 150 bp paired-end protocol. Raw data from the Sequencer were processed using Illumina’s Real-Time Analysis (RRID: SCR_014332) and CASAVA pipeline software version 1.8.2 (RRID:SCR_001802) by IGS, which includes image analysis, base calling, sequence quality scoring, and index demultiplexing. Data were then processed through both FastQC version 0.12.0 (RRID:SCR_005539) and in-house pipelines for sequence assessment and quality control. We assessed basecall quality, and reads with a median Phred-like quality score of lower than Q20 were truncated.

Sequencing read alignment and quantification: We used STAR aligner version 2.4.2a [15], with the default parameters, to map the short sequencing reads onto the Genome Reference Consortium Mouse Build 39 (GRCm39) [16]. We used mouse gene annotations from Ensembl Release 100 [17] for the mapping. RSEM version 1.2.22 [18] was used for the quantification of gene expression in Transcripts Per Million (TPM) at both gene and isoform levels. We used DESeq2 version 1.42.0 [19] for the differential expression analysis between the SCI and sham samples. R “stats” package version 4.3.2 was applied for the k-means clustering. We used clusterProfiler 4.10.0 [20] for the Gene Ontology analysis. The analysis of *Ntrk2* (trkB) splice variants were performed as in Pattwell et al. [21] using the RSEM-produced isoform-level TPM values. The full length and truncated isoforms of *Ntrk2* gene have Ensembl transcript ID ENSMUST00000079828 and ENSMUST00000109838, respectively. Non-major isoforms (median TPM < 5) are not included in the figure.

### 2.4. Statistical Analysis

All quantitative data are presented as individual data points in column graphs with mean ± SEM. Statistical analysis was performed on Sigma-Plot, version 12 (Systat Software; RRID: SCR_003210) for BMS scores and Graphpad Prism Version 9.5.0 for Windows (Graphpad Software, RRID: SCR_002798) for all other tests. For normality assessment of data distribution, the Shapiro–Wilk test was used.

For multiple comparisons between groups, one-way ANOVA analyses were performed followed by Tukey’s or Newman–Keuls multiple comparisons post hoc test for parametric (normality and equal variance passed) data. BMS scores were analyzed with two-way ANOVA for repeated measurements, followed by Holm–Sidak’s post hoc test for multiple comparisons. Statistical analysis in each assay was detailed in figure legends and a *p*-value ≤ 0.05 was considered statistically significant.

Multivariate data analysis was used to gain a comprehensive understanding of all the behavior tests in the three strains after mSCI. The behavior omics data included BMS, CatWalk, TS, FS, von Frey filament, and hot plate. The partial least squares discriminant analysis (PLS-DA) was conducted to classify mice behaviors based on the group effect of strain or injury using self-coded R language based on the ropls and mixOmics (version 6.20.0) packages [22].

## 3. Results

### 3.1. Spontaneous Motor Activity and Anxiety-like Behavior Vary across Three Mouse Strains

To assess baseline locomotion differences between the three mouse strains, the open field (OF) test was used in these naïve mice. Baseline motor function was significantly lower in both albino mice strains, A/J and BALB/c, compared to C57BL/6 mice (Figure 1). In the test period of 5 min, C57BL/6 showed the highest total distance travelled (*p* < 0.001, n = 25 mice/group, Figure 1A), while no differences were observed between A/J and BALB/c mice. Moreover, the motor function of albino mice is also reflected in movement speed, with both mean (*p* < 0.0001, Figure 1B) and max (*p* < 0.0001, Figure 1C) movement speeds in the OF test being significantly lower than mice from the C57BL/6 strain. In addition to gross motor function, the OF test has also been widely used to assess anxiety-like behavior in animals [23]. Thus, we analyzed the time each mouse spent in the inside and outside zones of the OF arena (Figure 1D,E). Compared to C57BL/6 mice, the two albino strains spent more time in the outside zone (*p* < 0.0001, Figure 1D), suggesting that the two strains may have inherent tendencies towards anxiety-like behavior. Taken together, our findings suggest that strain differences occur in naïve mice.

### 3.2. SCI Causes Locomotor Functional Deficits in Three Mouse Strains in an Injury-Severity-Dependent Manner

To assess how differing genetic backgrounds affect functional recovery following SCI, we subjected young adult mice to either a moderate/severe or mild injury at the T10 spinal cord. Mice were tested for BMS scores on day 1, day 3, and weekly thereafter for up to 8 weeks post-injury. BMS scores analysis showed that significant genotypes were observed between the three strains. Statistical analysis with two-way repeated measurements ANOVA showed significant main effects in strain (F (1,39) = 9.507, *p* < 0.001, Figure 2A). Starting at 2 weeks (w) following a moderate/severe contusion-based SCI, C57BL/6 mice showed better recovery than the A/J strain (*p* = 0.004), with an average score of 2.9, indicating that most C57BL/6 mice exhibited extensive ankle movement and plantar paw placement. The differences in hindlimb motor function became even more prominent in the weeks thereafter (*p* < 0.001, Figure 2A), as the A/J strain continued to exhibit an average score of less than 2, indicating that the majority of mice only showed slight ankle movement. The other albino strain we tested, BALB/c, also demonstrated faster endogenous recovery than A/J mice, showing an average score of 2.8 and significant differences in statistical analysis (*p* = 0.01).

Due to the lack of endogenous recovery in the A/J mouse strain, we reasoned that a moderate/severe SCI may have been too high in injury severity. In addition to affecting the results of BMS scoring, the lack of plantar paw placement and stepping in A/J mice was an impediment for further neurological assessment of fine motor function and pain sensitivity. To this end, we subjected a separate set of mice to mild SCI and evaluated their recovery progress through weekly BMS. Overall, significant main effects of strain (F (1,33) = 8.829, *p* < 0.001, Figure 2B) were also observed in this second set. Moreover, strain differences were significant starting as early as 1 d post-injury, with C57BL/6 showing higher scores than either BALB/c (*p* < 0.001) or A/J (*p* < 0.001) strains. This high level of hindlimb motor function recovery remained in the weeks thereafter, as the scores for C57BL/6 mice plateaued around 8 in week 4, indicating an almost full recovery with slight deficits in fine motor coordination. In contrast, mice of the BALB/c and A/J strains were only able to reach a maximum score of 7.75 and 6.79, respectively, suggesting high prevalence of hindlimb rotation and trunk instability. In general, the overall trends of motor function recovery remained the same in both moderate/severe and mild-SCI groups, with C57BL/6 showing the fastest progress and the highest extent of recovery.

Next, we used Catwalk automated gait analysis to detect potential motor differences beyond that recognized with BMS scores in mild-SCI groups where animals can place the plantar surface of the hindpaws on a flat surface. For evaluation of motor coordination, we used a regularity index, a parameter that tracks the order of paw placement in a step cycle (Figure 2C). A single step cycle is defined as each of the four paws being placed on the walking surface in sequence. Through analysis of step sequencing, the program then attributes each set of steps into either a normal stepping pattern or abnormal gait, with the result being a percentage of normal stepping out of all step cycles analyzed. Ideally, the regularity index of healthy C57BL/6 mice would be 100% [10,24]; though in practice, the results vary. Whereas, both C57BL/6 and BALB/c mice had a regularity index of 80 to 90% under normal conditions, A/J mice had an index score close to 50%, indicating baseline deficits in motor coordination. Following SCI, the regularity index of neither C57BL/6 nor A/J showed significant differences between sham and injury groups, while only BALB/c mice showed a significant injury effect (*p* < 0.05, Figure 2C). Print position is defined as the distance between a pair of hindpaw and forepaw of the same side. Ideally, healthy C57BL/6 mice should be able to place their hindpaw next to the location of the forepaw that has just been lifted from the walkway. After a mild contusion, the print position of A/J mice showed no SCI-mediated differences, which is likely due to the already prominent deficits observed in the sham group (*p* < 0.001 vs. BALB/c/Sham, *p* < 0.0001 vs. C57BL/6/Sham, Figure 2D). For C57BL/6 and BALB/c mice, however, print position was dramatically increased in injured mice (*p* < 0.05). Stride length, which is defined as the total distance between steps of the same paw, was significantly lower in A/J mice compared to BALB/c and C57BL/6 strains, regardless of injury. However, no injury effects were observable in any of the three strains for this parameter (Figure 2E). Hindlimb base-of-support is a parameter that measures the average width of the track (distance between RH and LH) made by the animal, in which the farther apart the feet are placed during locomotion, the less likely the animal is to fall and the larger the base-of-support [24]. Strain differences in base-of-support were observed between C57BL/6 and BALB/c in sham mice (*p* < 0.001, Figure 2F), with C57BL/6 showing a smaller track width. After injury, C57BL/6 mice showed a dramatic increase in base-of-support (*p* < 0.01, Figure 2F), suggesting that mice showed compensation for instable gait. As an extension for gait stability assessment, we examined weight distribution, which calculates how many feet are simultaneously on the ground during a complete stride cycle. Injury effects were prominent in BALB/c mice, which saw a significant decrease in time spent in a diagonal stance (*p* < 0.01, Figure 2G), but no differences were seen in the time spent on girdle or four-paw stances. For A/J mice, significant SCI-induced differences were seen in time spent on a girdle stance (*p* < 0.0001, Figure 2H) and a four-paw stance (*p* < 0.01, Figure 2I), suggesting a redistribution of weight to compensate for motor deficits following injury. Taken together, the recovery process following SCI is significantly affected by the genetic background of mice strains, with C57BL/6 showing faster recovery and less deficits in overall motor function and coordination.

### 3.3. SCI-Mediated Depression-like Behaviors Show Strain Differences

When the BMS scores reached the plateau, a battery of neuropsychiatric behaviors was applied to those animals. To further explore the effects of genetic background on the prevalence of SCI-induced depression, we ran learned helplessness tests on the two sets of mice subjected to SCI. In the moderate/severe injured groups, mice from all three strains showed increased immobility time in the tail suspension (TS) test following injury (*p* < 0.01 for A/J and C57BL/6, *p* < 0.001 for BALB/c mice, Figure 3A), but no strain differences were observed in either the sham or SCI groups. Similarly, the immobility time was also increased in the forced swim (FS) test after SCI (*p* < 0.001 for A/J and C57BL/6, *p* < 0.01 for BALB/c mice, Figure 3B), with no strain differences detected. However, in mice subjected to mild SCI, the test results show dramatic differences depending on genetic background. The TS test showed higher immobility time in BALB/c mice following SCI (*p* < 0.05, Figure 3C), but not in A/J or C57BL/6 mice. However, strain differences were detectable when comparing the SCI mice of A/J to BALB/c (*p* < 0.01, Figure 3C) and C57BL/6 (*p* < 0.01). In the FS test, SCI-induced depression-like behavior was observed in both A/J (*p* < 0.0001) and BALB/c (*p* < 0.01) mice (Figure 3D). Like the TS test, dramatic strain differences were detected between the SCI mice of the A/J strain compared to either BALB/c (*p* < 0.01) or C57BL/6 (*p* < 0.05) mice. In contrast to moderate/severe SCI, C57BL/6 mice following mild SCI did not show significant depression-like behaviors. In the FS test, we also observed dramatic strain differences between the sham groups of each strain, suggesting varying degrees of susceptibility to depression. Taken together, our data suggest that genetic background and injury severity are the two crucial factors involved in the development of depression-like behaviors following SCI.

### 3.4. Chronic Mild SCI Elicits Cutaneous Hypersensitivity and Spontaneous Pain at the Hindpaws

To test whether genetic background is involved in the development of post-injury allodynia evoked by mechanical and thermal stimuli, we conducted nocifensive behavioral testing in mild SCI groups, in which animals were capable of placing the plantar surface of the hindpaws on a flat surface. In the von Frey test for mechanical allodynia, the withdrawal threshold of all three mice strains were significantly lower at 5 weeks post-injury compared to their respective sham groups (*p* < 0.01 A/J and C57BL/6, *p* < 0.05 in BALB/c, Figure 4A). However, when comparing strains, no differences were found for the mechanical threshold. The thermal threshold obtained via an incremental hot plate test showed no injury-induced hyperalgesia in A/J and BALB/c mice, but a dramatic decrease in C57BL/6 mice at 6 weeks post-injury (*p* < 0.01, Figure 4B) was observed. Additionally, there is a significant strain difference in both the sham and SCI groups, with C57BL/6 mice having a significantly lower threshold than the two albino strains.

Next, pain sensitivity was further explored with relevant parameters obtained from Catwalk gait analysis, which reflects spontaneous pain-type behaviors. Maximum contact area was defined as the total area of the hindpaw at the point of maximum placement, which showed no injury effects in either A/J or BALB/c mice at 8 weeks post-injury (Figure 4C). However, strain differences between sham groups should be noted, with C57BL/6 mice showing the highest contact area and the A/J mice showing the lowest. Following SCI, C57BL/6 mice showed a dramatic decrease in max contact area, which suggests the development of spontaneous pain. In tandem with this parameter was the mean print area, which also saw a dramatic decrease in C57BL/6 mice after SCI (*p* < 0.01, Figure 4D), but no injury effects in either A/J or BALB/c mice were observed. Similarly, strain differences were also observed between both the sham and SCI groups, suggesting fundamental variations in weight-bearing and gait patterns based on genetic background.

To collectively delineate the total behavioral profiles of the animals from different strains and injured groups, all the behavioral data from the mild-SCI set of the mice were applied for a machine learning approach, partial least squares discriminant analysis (PLS-DA) [25]. Remarkably, we observed that mice from each strain are tightly clustered in the result (Figure 5). This indicates that the genotypes of the three strains are highly correlated with their behaviors of locomotion, depression, and neuropathic pain.

### 3.5. Tissue Bulk RNAseq Analysis Reveals Differential Gene Expression in Three Mouse Strains after SCI

To examine the genomics between mouse strains following SCI, after completion of the behavioral tests at 8 weeks post-injury, bulk RNAseq from sham or injured spinal cord tissues was employed after moderate/severe SCI. Gene expression changes were analyzed with DEseq2 following normalization [19]. Principal component analysis (PCA) of all normalized gene counts revealed a clear separation of sham and SCI samples into individual groups across the first two principal components (PC, Appendix A). PC1 was the major principal component with a contribution of 67.59% of gene variations separating injury groups from sham groups, which may represent injury effects. PC2 contained 17.4% of gene variations and separated different strains of mice from sham or injured groups, which may represent strain effects. Differentially expressed genes (DEGs) with a Benjamini–Hochberg adjusted *p*-value of less than 0.05 in the Wald test were used for downstream analysis. DEGs between the SCI mice and sham group of each strain were presented in the heatmap for visualization after k-means clustering (Figure 6A), with each column representing the fold difference in log2 scale. Based on the heatmap results, we were able to identify 11 clusters of DEGs for further pathway enrichment analysis with the gene ontology (GO) biological process database. GO analysis of cluster 1, which consisted of DEGs downregulated in all the three strains by SCI, revealed genes involved with the regulation of membrane potential as the top enriched pathway (Figure 6B). Other pathways indicated for cluster 1 include learning or memory, dendrite development, and regulation of synapse activity or organization. These pathways suggest downregulation of genes contributing to the normal homeostasis of membrane potential and synaptic function after SCI. In contrast, the upregulated genes that comprised cluster 6 were mostly involved with the regulation of immune effector and innate immune response (Figure 6C), in which DEGs overlapped with more than 10% of the genes in this functional category. Of the other top enriched GO terms, leukocyte proliferation, regulation of response to biotic simulations, and leukocyte cell adhesion were in the top five. The GO terms for this cluster indicate the presence of persistent neuroinflammation in the injured spinal cord for all three strains. Consistently, GO analysis of cluster 5 showed highly upregulated immune genes in response to injury in all three mouse strains (Appendix A). However, cluster 2 showed upregulated genes in the other strains, but not in A/J mice (Appendix A). Of note, some genes were downregulated most prominently in A/J or BALB/c strains, which can be observed in cluster 11 (Figure 6D) and cluster 9 (Appendix A). GO analysis of these clusters showed DEGs were highly enriched with genes regulating the cholesterol metabolic process and muscle organ development. Other functions implicated via GO analysis include 2nd alcohol metabolic process, sterol metabolic process, and potassium (K+) ion transmembrane transport and K+ ion transport. Cluster 7 represented downregulated genes in all three mouse strains after SCI (Appendix A). The genes in this cluster are about 50% more downregulated in C57BL/6 mice.

Delving deeper into the specific genes of each cluster, we found significant upregulation of inflammatory genes (Figure 6E). DEGs of interest include *Atf3* (activating transcription factor 3), a member of the mammalian activation transcription factor/cAMP responsive element-binding (CREB) family and a gene involved with cellular stress response and neuronal damage [26]. Moreover, *Ccl2* (C-C Motif Chemokine Ligand 2) and its corresponding receptor *Ccr2* (C-C Motif Chemokine Receptor 2) were upregulated, both of which are involved with the development of nociceptive behavior following SCI [27]. An interesting phenomenon is the downregulation of *Il6* (Interleukin 6) in C57BL/6 mice, which studies found are crucial to the development of neuropathic pain following SCI [28,29], whereas it was upregulated in both A/J and BALB/c mice. In addition, the proinflammatory cytokines *Il1b* (Interleukin 1 beta) and *Tnf* (Tumor necrosis factor) were upregulated in all three strains. Furthermore, the genes *Prkcd* (Protein Kinase C Delta) and *Ptgs2* (Prostaglandin-Endoperoxide Synthase 2), were upregulated more prominently in A/J mice compared to other strains. In order to explore the effects that differing genetic backgrounds may have on classic receptors involved with neuropathic pain, we examined the expression changes in TRP (transient receptor potential) receptors (Figure 6F). All three genes examined (*Trpa1*, *Trpm1*, and *Trpv1*) were upregulated more significantly in A/J mice following injury compared to BALB/c and C57BL/6 mice. Notably, *Trpa1* (Transient Receptor Potential Cation Channel Subfamily A Member 1), a gene that is involved in familial episodic pain syndrome [30] and other forms of nociceptive behavior [31,32,33], was highly upregulated in A/J mice, but only minor changes were induced by SCI in the other strains. In contrast, all DEGs from the NMDA/GABA receptor family were found in clusters 1 and 4, showing significant downregulations after SCI (Figure 6G). All DEGs from the potassium channels were found in cluster 11, showing significant downregulations following SCI in all three mouse strains (Figure 6H).

Moreover, we found significant upregulation of cell cycle genes after SCI in all three mouse strains (Figure 7A), which includes *Ccnd1* (cyclin D1), *Ccnd2* (cyclin D2), and *Cdk4*. Strain differences were most prominent in *Ccnd2*, known to be involved in secondary injury damage following SCI [6]. In agreement with our previous studies [9,10], SCI significantly upregulated a truncated isoform of *Ntrk2* (trkB.T1) but downregulated the full length of trkB (Figure 7B). Strain differences were most prominent in *Ntrk2*, known to be involved in SCI-PAIN [9,10]. Together, our data demonstrate differential gene expression in three mouse strains after chronic SCI which is associated with genetic background.

## 4. Discussion

Although pre-clinical studies of SCI suggest that genetic factors significantly impact behavioral recovery [1,34,35], their relationship to recovery patterns across different genetic mouse strains has been little studied. Here, we evaluated behavioral changes in locomotion, allodynia, and depression across several strains of mice commonly used in SCI research and used RNAseq to dissect associated differential gene expression and pathway regulation. Our findings demonstrate that experimental SCI causes different profiles of functional deficits across three mouse strains and at different injury-severity levels. SCI-mediated depression-like behaviors and cutaneous or spontaneous hypersensitivity in the hindpaws showed robust strain differences. There were also altered motor activity and anxiety-like behaviors identified across the strains in naive animals. We also demonstrated significantly different gene expression and pathway-regulation patterns in the three mouse strains after chronic injury.

Strain differences have been extensively reported with regard to motor behaviors in naive mice [36,37]. In agreement with previous reports [36,37,38], both A/J and BALB/c mice show lower spontaneous motor activity, but higher levels of anxiety-like behavior than C57BL/6 animals, indicating that genetic background plays a crucial role for these behaviors. Following contusive injury in the mid thoracic spinal cord, mice of different strains (C57BL/6, C57BL/10, B10.PL, BALB/c, and C57BL/6x129S6 F1 strain) showed differing levels of locomotor recovery tested at different injury-severity levels [1]. A/J mice appeared more susceptible to SCI and had more severe persistent locomotor deficits than the other strains. Although BALB/c mice had similarly low open field activity, their endogenous locomotor recovery following SCI was better than that in the A/J strain. Among these three mouse strains, C57BL/6 mice showed the highest BMS scores after SCI. Mice of the A/J and BALB/c strains are albino with different brain anatomic structure [39,40,41,42]. Following unilateral sciatic nerve crush, the A/J strain showed elevated expression of major histocompatibility complex class I and increased microglial reaction and astrogliosis compared to those in C57BL/6 animals [43]. BALB/c mice were significantly more susceptible to tissue damage resulting from permanent focal cerebral ischemia than C57BL/6 mice [44]. In a contusion SCI model, Kigerl et al. [45]. reported varied glial-related inflammatory responses to injury between strains.

Utilizing the Catwalk XT for detecting sensitive motor characteristics, we observed significant strain differences in motor coordination, as well as general motor function in step cycles. The A/J mice showed low levels of sensorimotor gating including step cycles and stride length, as well as abnormal print position and weight distribution. After a mild contusion injury, the A/J strain showed no SCI-mediated differences, which may reflect the worse motor coordination observed in their sham controls. The BALB/c mice revealed abnormal gait stability in response to mild SCI. Thus, both albino strains, BALB/c and A/J, have a slower recovery process and deficits to fine motor coordination which are more likely to be detected through Catwalk gait analysis. These data are consistent with the conclusion that genetic differences across mouse strains may underlie different post-traumatic deficit levels after SCI [42].

Most post-SCI pain states are presumed to be secondary to soft-tissue and nerve damage, but only a subset of the population suffers from chronic pain after SCI [46]. Therefore, it is believed that chronic SCI pain requires additional factors beyond pre-existing co-morbidities, such as genetic factors [3]. It is not difficult to demonstrate the variability of acute pain responses in laboratory settings; standardized noxious heat stimuli elicit pain ratings near zero in some individuals and near maximal (‘worst pain imaginable’) in others [47]. With chronic pain after SCI, the variability of severity, symptoms, and presentation of that pain is not predictable based on severity of injury alone [48]. Here we demonstrated differential changes in allodynia in three inbred mouse strains after mild SCI, a model known to cause allodynia [10], confirming genotype dependence of the development of mechanical allodynia. In assessment of cutaneous thermal hypersensitivity and spontaneous pain in the hindpaws, the C57BL/6 strain showed a high sensitivity response to injury, in contrast to A/J and BALB/c mice. Overall, our results indicate that pain sensitivity to mechanical/thermal stimulus, as well as spontaneous pain-like behavior, differ markedly across mice with different genetic backgrounds.

Co-morbid secondary complications are also known to arise as a result of SCI. In addition to the physical limitations caused by the initial injury, pain and depression can occur and have been linked to higher morbidity, mortality, suicide, substance abuse, and deleterious effects on physical and social functionality [49]. After SCI, the incidence rate of depression is more than double, increasing to between 25% and 47% [50,51]. It is not yet known if genetic background affects the prevalence of SCI-induced depression. In the present study, we showed that, in response to moderate/severe SCI, mice from all three strains showed depressive-like behaviors assessed with forced swim and tail suspension tests, two known learned-helplessness tests. However, in mice subjected to mild SCI, both albino strains showed elevated immobility time in these tests, suggesting increased depression, in contrast to C57BL/6 mice. This may reflect, in part, the greater locomotor deficits in the albino strains. In addition, A/J mice have low magnesium-ion levels in both plasma and brain tissue compared to the C57BL/6 strain, which is known to be correlated with several anxiety-related behavioral parameters [52]. The BALB/c mice, a strain exhibiting high susceptibility to stress, are recognized as being “anxiety-prone” or “highly anxious” compared to other strains of mice [53,54]. This strain is also considered to be a useful experimental model for depression.

Although the application of genetics to pain is in its infancy compared to many other fields (e.g., hypertension, diabetes, and drug abuse), significant strides have been made in the last two decades [3,55]. A growing number of quantitative trait loci associated with variable pain sensitivity among inbred mouse strains have been discovered. However, less attention has been paid to comparative genetics and genomics between mouse strains following SCI. In the present study, bulk RNAseq analysis from sham or injured spinal cord tissue revealed differential gene expression across the three mouse strains after chronic SCI. The genes that are commonly upregulated after SCI include inflammatory genes and TRP (transient receptor potential) receptors in the present study. Among these, the pro-inflammatory genes *Il1b*, *Tnf*, and *Ptgs2*, and all *Trp* receptors genes, were upregulated more prominently in A/J mice compared to other strains. In contrast, both *Il6* and *Trpa1* genes, crucial to the development of neuropathic pain following SCI [28,29,31,32,33], were downregulated in the C57BL/6 mice. SCI-mediated downregulated genes include the NMDA/GABA receptor family and the potassium channel genes. Of the three GABA receptors examined, strain differences were most prominent for *Gabrg1* (Gamma-Aminobutyric Acid Type A Receptor Subunit Gamma1), which has been implicated in epilepsy [56,57] and alcoholism [58,59], as well as with depression and suicide behavior [60]. Among the potassium channels genes, strain differences were most prominent in *Kcnv2* (Potassium Voltage-Gated Channel Modifier Subfamily V Member 2) and *Lrrc26* (Leucine Rich Repeat Containing 26), which has been implicated in chronic neuropathic pain [61].

Our previous comprehensive gene profiling analyses of the rat spinal cord after contusion injury have demonstrated upregulation of a cluster of cell cycle-related genes [6,7], suggesting involvement of cell cycle-related genes in neuronal damage and subsequent cell death. We also demonstrated, by using microarray analysis, the existence of a delayed expression cluster of inflammation-related genes that may play a role in secondary injury [8]. In agreement with these reports, several key cell cycle genes were upregulated after SCI in all three mouse strains, but the A/J mice showed the least increase in the *Ccnd2* gene. More recently, to examine the mechanisms underlying the decrease in nocifensive behavior in mice following SCI in the absence of trkB.T1, we conducted a microarray experiment to test differential gene expression in transcriptional pathways that control SCI-PAIN. We demonstrated that the regulation of cell cycle genes was altered in mutant mice [9,10]. These studies demonstrate the powerful potential of differential gene expression to provide mechanistic insights into the development and/or persistence of SCI-PAIN. In the present study, SCI in both A/J and BALB/c mice significantly upregulated a truncated isoform of *Ntrk2* (trkB.T1) but downregulated the full length of *trkB* in all three strains. This contrasts with the previous study which found that the C57BL/6 strain exhibited high *Ntrk2* but an unchanged full length of *TrkB*, possibly due to injury severity [9].

## 5. Conclusions

The present study highlights the importance of genetic variants in the assessment of functional outcome following experimental SCI models. Mice of different strains subjected to similar SCI demonstrate significant differences in locomotor recovery and anxiety/depression-like behaviors, and mice develop significant tactile allodynia, which is associated with differential gene expression and pathway regulation. Table 1 summarized the results of the various tests in A/J, BALB/c, and C57BL/6 male mice. These findings support the concept that genetic and genomic changes alter the trajectory of SCI behavioral responses. As the use of genetically engineered animal models to explore SCI pathobiology becomes increasingly common, there is a heightened need for careful interpretation of behavioral outcomes. Our findings indicate that such differences are likely confounders in studies using genetically modified mice.

## Figures and Tables

**Figure 1 cells-13-00759-f001:**
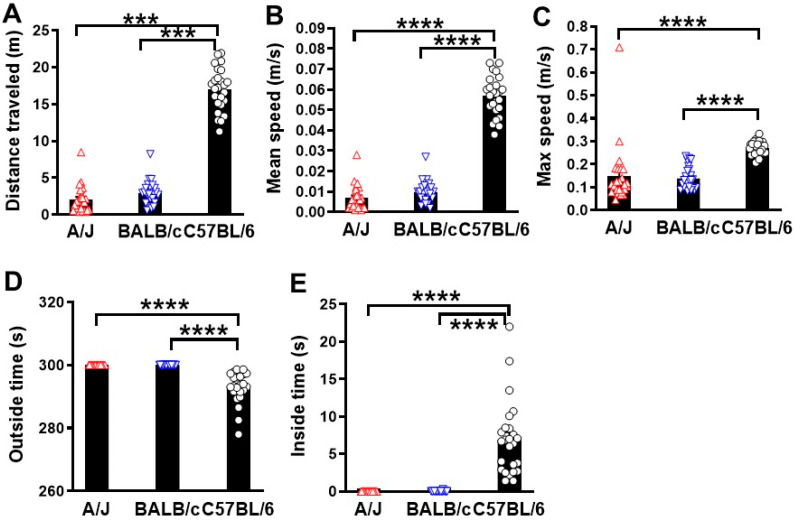
Spontaneous motor activity and anxiety-like behavior are altered in three mouse strains (A/J, BALB/c, and C57BL/6). Distance traveled (**A**), mean speed (**B**), max speed (**C**), outside times (**D**), and inside times (**E**) are listed. n = 25 mice/group. *** *p* < 0.001, **** *p* < 0.0001. One-way ANOVA followed by Tukey’s multiple comparisons test.

**Figure 2 cells-13-00759-f002:**
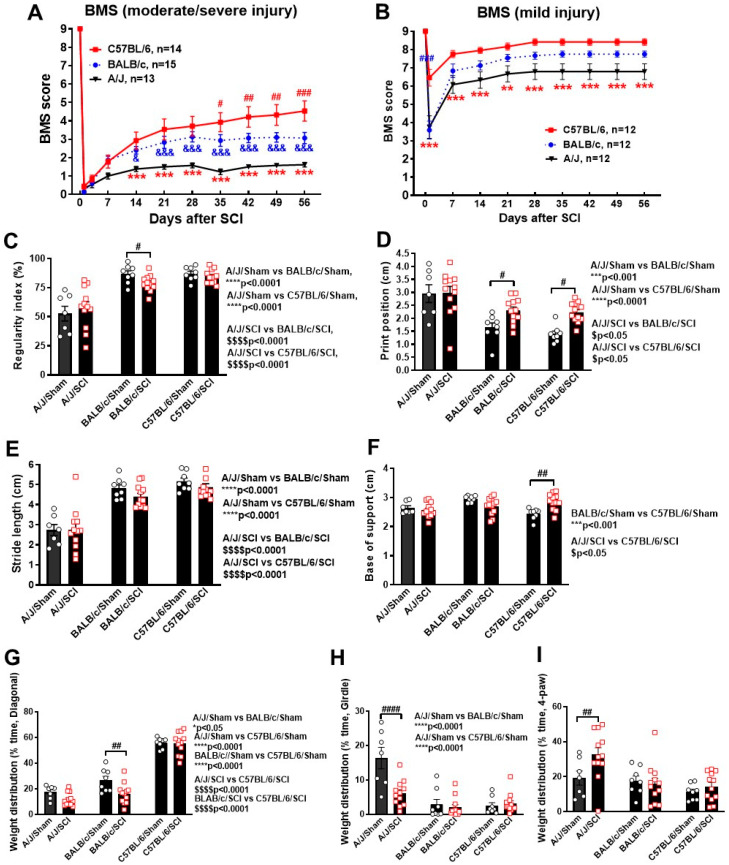
Spinal cord injury induces differential locomotor function and motor coordination in three mouse strains. (**A**,**B**) BMS scores of all three strains were recorded weekly to quantify the recovery process of hindlimb locomotor function following a moderate/severe (**A**) or mild (**B**) contusion injury. n = 10 (Sham) and 13–15 (moderate/severe SCI), 7–8 (Sham), and 12 (mild SCI) mice/group. ** *p* < 0.01, *** *p* < 0.001 vs. C57BL/6 group; & *p* < 0.05, &&& *p* < 0.001 vs. BALB/c mice; # *p* < 0.05, ## *p* < 0.01, ### *p* < 0.001 vs. BALB/c group. Two-way ANOVA with repeated measurement followed by Holm–Sidak’s multiple comparisons test. (**C**–**I**) Catwalk XT automated gait analysis was used to evaluate differences in motor coordination at 8 weeks post-injury for the mild-SCI groups. The parameters tested include regularity index (**C**), print position (**D**), stride length (**E**), base-of-support (**F**), and time spent in the weight distribution stance pattern of diagonal (**G**), girdle (**H**) and four-paws (**I**). n = 7–8 (Sham) and 12 (mild SCI) mice/group. # *p* < 0.05, ## *p* < 0.01, #### *p* < 0.0001 vs. their sham groups. One-way ANOVA followed by Newman–Keuls multiple comparisons test.

**Figure 3 cells-13-00759-f003:**
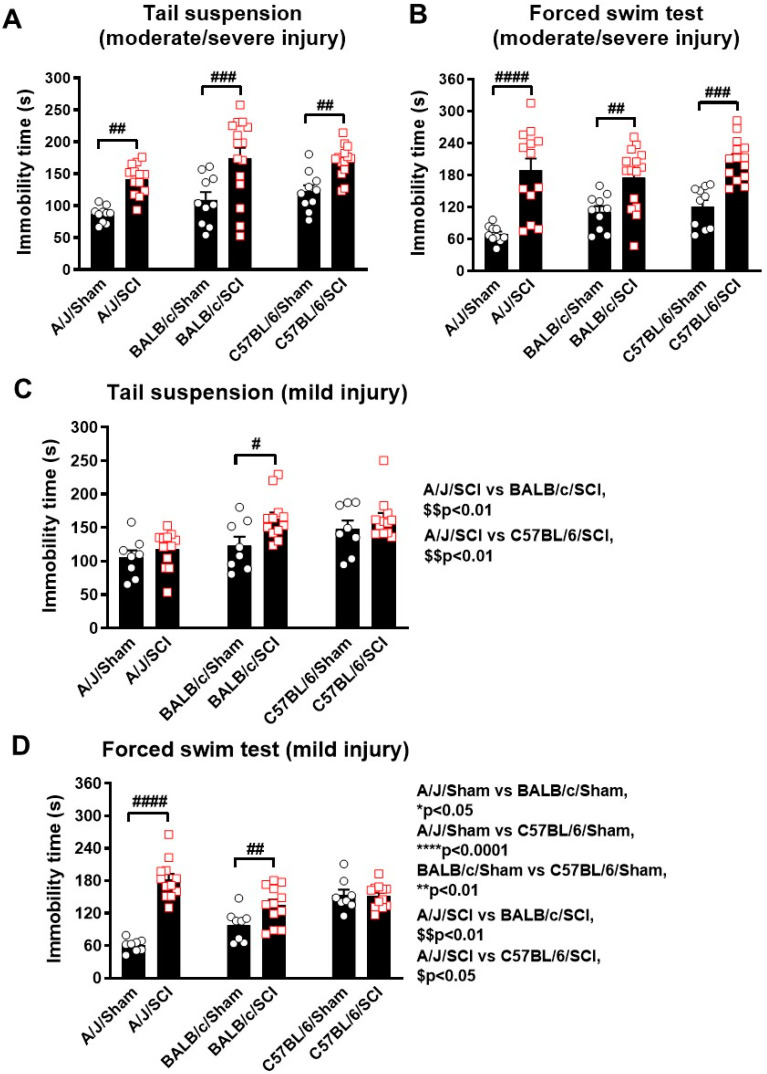
Spinal cord injury induces differential depression-like behaviors in three mouse strains at 7–8 weeks post-injury. (**A**,**B**) Depression-like behaviors were assessed with tail suspension (A, TS) and forced swim (B, FS) tests following a moderate/severe contusion injury. SCI-induced immobility time was significantly increased in all three strains. n = 10 (Sham) and 13–15 (moderate/severe SCI) mice/group. (**C**,**D**) In mild-SCI groups, depression-like behaviors were examined with TS (**C**) and FS (**D**) tests. n = 7–8 (Sham) and 12 (mild SCI) mice/group. # *p* < 0.05, ## *p* < 0.01, ### *p* < 0.001, #### *p* < 0.0001 vs. their sham groups. One-way ANOVA followed by Newman–Keuls multiple comparisons test.

**Figure 4 cells-13-00759-f004:**
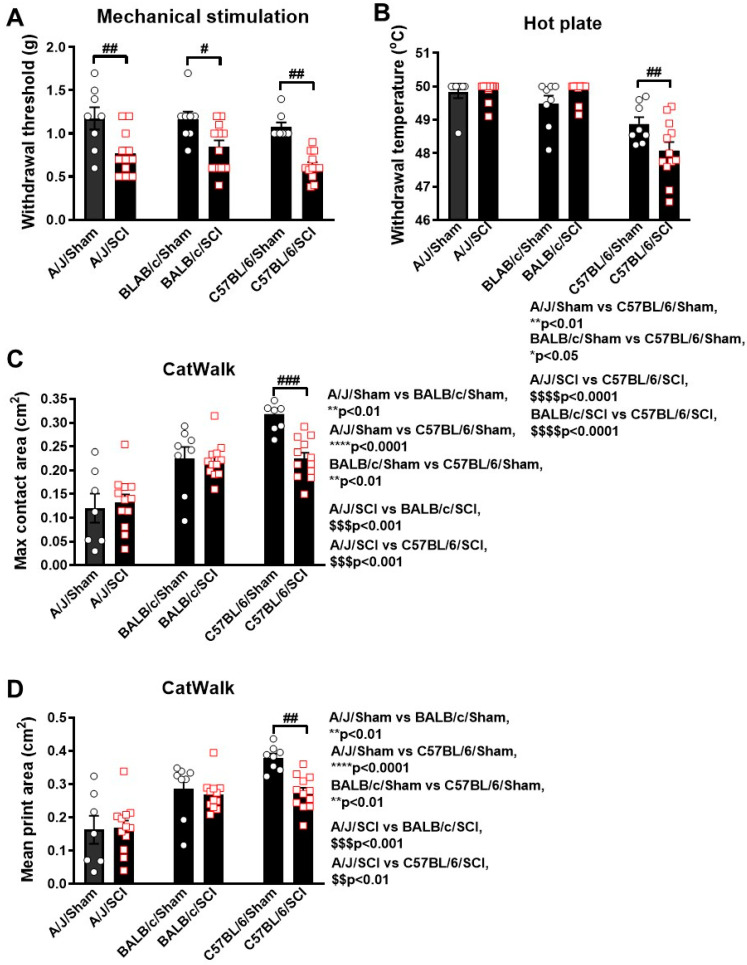
Behaviors indicative of neuropathic pain show differing sensitivity in three mouse strains following mild spinal cord injury. (**A**) Mechanical allodynia was detected using von Frey filaments, showing an injury-induced hypersensitivity to mechanical stimulation in all three strains at 5 weeks post-injury. (**B**) A hot plate test was utilized to detect thermal hyperalgesia, which was only detectable in C57BL/6 mice at 6 weeks post-injury. (**C**,**D**) Parameters extracted from Catwalk gait analysis that reflect pain sensitivity include max contact area (**C**) and mean print area (**D**) at 8 weeks post-injury. n = 7–8 (Sham) and 12 (mild SCI) mice/group. # *p* < 0.05, ## *p* < 0.01, ### *p* < 0.001 vs. their sham groups. One-way ANOVA followed by Newman–Keuls multiple comparisons test.

**Figure 5 cells-13-00759-f005:**
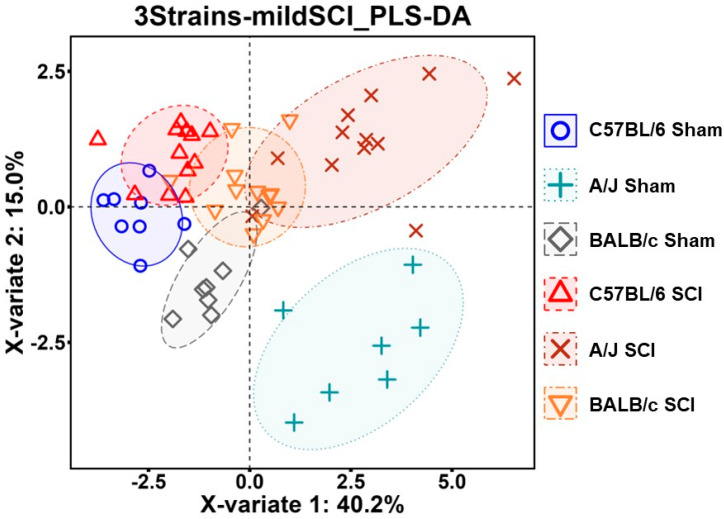
Multivariate data analysis of the behavior omics in three mouse strains after mild SCI. Integrated behavioral data were applied for PLS-DA as clusters by group. n = 7–8 (Sham) and 12 (mild SCI) mice/group.

**Figure 6 cells-13-00759-f006:**
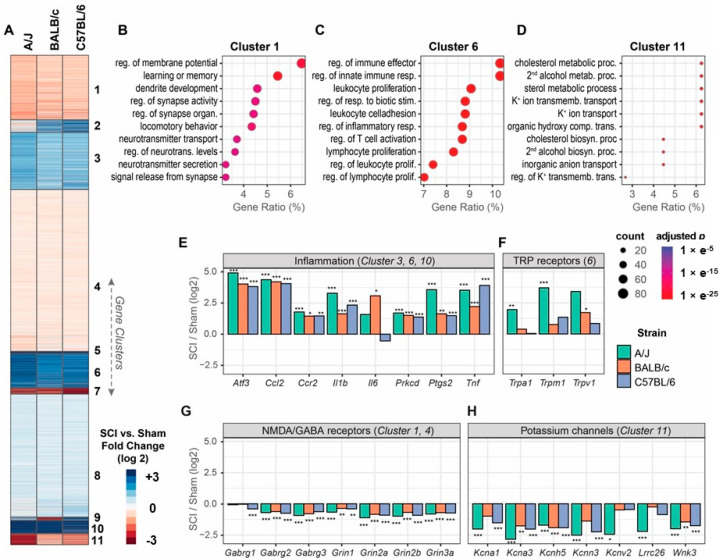
Tissue bulk RNAseq analysis showed differential gene expression in three mouse strains at 8 weeks after moderate/severe SCI. (**A**) Gene expression changes are summarized in the heatmap. Each column of the heatmap represents the fold difference in gene expression between the SCI mice and sham groups in log2 scale. Genes (rows) are clustered using the k-means clustering method. Genes whose change is not significant (adjusted *p* > 0.05) in all of the comparisons are not included. (**B**–**D**) Gene ontology (GO) analysis showed the results for cluster 1, 6, and 11, respectively. Dot sizes indicate the number of genes with the enriched GO terms, and colors indicate the significance levels. Gene ratio on *x*-axis indicates % of genes in a GO term compared to the total number of genes in that category. GO terms are abbreviated for brevity. (**E**–**H**) Examples of genes that are commonly upregulated (**E**,**F**) and downregulated (**G**,**H**) in the clusters in (**A**). n = 4 mice/group. * adjusted *p* < 0.05, ** adjusted *p* < 0.01, *** adjusted *p* < 0.001 in Wald test corrected with Benjamini–Hochberg method.

**Figure 7 cells-13-00759-f007:**
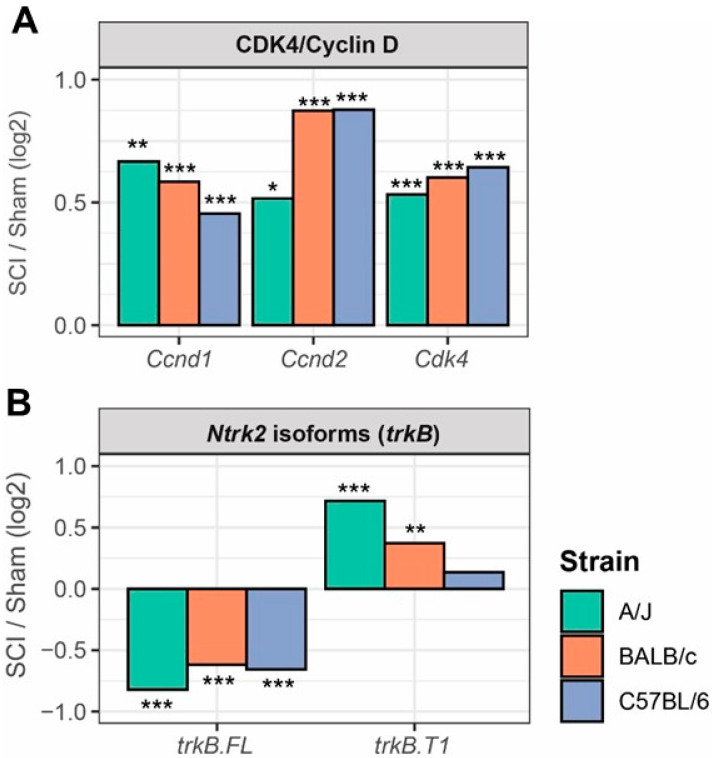
SCI in three mouse strains mediates differential gene expression in cell cycle pathway and the brain-derived neurotrophic factor (BDNF) receptor—tropomyosin related kinase B (trkB) gene Ntrk2. (**A**) SCI induces significant upregulation of cell cycle genes in three mouse strains, which includes *Ccnd1* (cyclin D1), *Ccnd2* (cyclin D2), and *Cdk4.* (**B**) SCI significantly upregulated a truncated isoform of *Ntrk2* (trkB.T1) but downregulated the full length of trkB. n = 4 mice/group. * adjusted *p* < 0.05, ** adjusted *p* < 0.01, *** adjusted *p* < 0.001 in Wald test corrected with Benjamini–Hochberg method.

**Table 1 cells-13-00759-t001:** Summary of the results of the various tests in A/J, BALB/c, and C57BL/6 male mice.

Mouse Strains	Locomotion	Affective/Depression	Nocifensive	Genomic	Cell Cycle	Ntrk2 Isoforms
**C57BL/6**	Spontaneous movement at baseline**↑** Recovery after SCI**↑** Motor coordination**↑**	Depression-like behaviors following mod/severe SCI**↑** but not affected by mild SCI**↔**	Sensitivity to mechanical and thermal stimulus after mild SCI**↑** Spontaneous pain**↑**	Inflammation**↑** Il6**↓** TRP receptors**↑** Trpa1**↔** NMDA/GABA**↓** Potassium channel**↓**	Ccnd1**↑** Ccnd2**↑↑** Cdk4**↑**	trkB.FL**↓** trkB.T1**↑**
**BALB/c**	Spontaneous movement at baseline**↓** Recovery after SCI**↓** Motor coordination**↓**	Depression-like behaviors after mod/severe**↑** and mild SCi**↑**	Sensitive to mechanical stimulus after mild SCI**↑**Thermal allodynia and spontaneous pain**↔**	Inflammation**↑** TRP receptors**↑** NMDA/GABA**↓** Potassium channel**↓**	Ccnd1**↑** Ccnd2**↑↑** Cdk4**↑**	trkB.FL**↓** trkB.T1**↑**
**A/J**	Spontaneous movement at baseline**↓** Recovery after SCI**↓↓** Motor coordination**↓**	Depression-like behaviors after mod/severe and mild SCi**↑** Immobility time in FS**↑** but not TS after mild SCI**↔**	Sensitive to mechanical stimulus after mild SCI**↑** no thermal allodynia & spontaneous pain**↔**	Inflammation**↑↑** TRP receptors**↑↑** NMDA/GABA**↓** Potassium channel**↓↓**	Ccnd1**↑** Ccnd2**↑** Cdk4**↑**	trkB.FL**↓** trkB.T1**↑↑**

SCI: spinal cord injury; FS: forced swim; TS: tail suspension; Il6: Interleukin 6; TRP: transient receptor potential; Trpa1: Transient Receptor Potential Cation Channel Subfamily A Member 1; Ccnd: cyclin D; Cdk4: cyclin kinase4; trkB.FL: tropomyosin related kinase B full length; trkB.T1: trkB truncated isoform (*Ntrk2*). Black arrows: baseline changes; Red arrows: SCI changes.

## Data Availability

All data needed to evaluate the conclusions in the paper are present in the paper and/or the Appendix A.

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
