# Peer review of "Dissecting Genetic Mechanisms of Differential Locomotion, Depression, and Allodynia after Spinal Cord Injury in Three Mouse Strains"

_cells, 2024, doi:10.3390/cells13090759_

Round 1

Reviewer 1 Report

Comments and Suggestions for Authors

The authors present a particularly interesting study comparing the effects of SCI on different strains of mice.

The study is particularly well conducted and presents functional data including locomotor tests, sensory tests and cognitive tests after SCI. The authors also present results on the spontaneous activity of these mice.

The authors then conducted RNASeq studies on these three mouse lines.

The data presented is therefore particularly rich and interesting.

If possible, I would like the authors to present a study of spinal scar after injury in these three strains, analysing in particular glial and fibrotic scars and inflammation.

In addition, from a presentation point of view, I think it would be interesting for the authors to present a table summarising the results of the various tests and the main differences found between these three strains of mouse. 

Author Response

If possible, I would like the authors to present a study of spinal scar after injury in these three strains, analysing glial and fibrotic scars and inflammation.

Response: We agree with the reviewer that histopathologic analysis of spinal cord tissue will provide more in depth at the contributions of genetics in three strains of mice after SCI. However, we apologize for the lack of relevant histological data. This project was conducted several years ago. Following the relocation of our lab, many histological and tissue samples for this project were lost in storage. So, we don’t have ready-to-use samples that can be stained for glia scarring and inflammation. Redoing the model and testing on a new batch of mice would require a strenuous effort, starting with amendments to our current IACUC protocol, ordering new mice from Jackson Laboratories, and up to 8 weeks of testing after injury. In total, it would take up to 6 months for additional data. Further histopathologic comparison in the mouse strains commonly used in experimental research will be intriguing for future studies.

In addition, from a presentation point of view, I think it would be interesting for the authors to present a table summarising the results of the various tests and the main differences found between these three strains of mouse. 

Response: We have added a table summarizing our main findings to table 1 of the revised manuscript. (See page 18).

Reviewer 2 Report

Comments and Suggestions for Authors

The manuscript entitled " Dissecting genetic mechanisms of differential locomotion, depression, and allodynia after spinal cord injury in three Mouse Strains" by Yang et al highlights  that the genetic background of the mouse strains significantly affects the recovery process following SCI, indicating a role of genetic mechanisms in differential locomotion after spinal cord injury. I have some suggestions that would help bolster the findings of the paper.

1. How was the outside and inside demarcated for the open field test. It needs to be mentioned in methods.

2. Behavioral assessment for the open field was done how many days after surgery? needs to be mentioned

3. What was the recovery time pos surgery? needs to be mentioned.

4. For anxiety like behaviors more than one test is needed such as marbel burying and fecal boli. Since open field is not the gold standard for anxiety behavior. It alone is not concrete evidence.

5. It would be beneficial to show the duration of nocifensive behavior over days until resolution rather than just one day.  If testing was done only for 1 day it would be better if that day is consistent between all behavior data. i.e von frey, hot plate and catwalk for day 6 instead of different days.

6. Why was 8 weeks chosen for RNA seq analysis? There was no explanation given, it seems out of the blue.

Author Response

  1. How was the outside and inside demarcated for the open field test. It needs to be mentioned in methods.

Response:  Per the reviewer’s suggestion, we have added details on the demarcation of outer/inner zone for to the methods section for open field test. On the Any-maze tracking software, the open field apparatus was divided into a grid of 10cm x 10cm squares. The outer 10cm zone was defined as the outer zone, whereas the center 20 x 20 cm2 region is defined as the inner zone. (pages 2-3 of the M&M section).

  1. Behavioral assessment for the open field was done how many days after surgery? needs to be mentioned

Response: The behavioral assessment for open field was performed on naïve mice before SCI. As written in the results section, these are baseline differences in locomotion. We will empathize this time point in the materials and methods section for clarification.  (page 5).

  1. What was the recovery time pos surgery? needs to be mentioned.

Response: The recovery time was up to 8 weeks after SCI, which was shown in the results for Basso mouse scale (BMS) scoring. (page 5).

  1. For anxiety like behaviors more than one test is needed such as marbel burying and fecal boli. Since open field is not the gold standard for anxiety behavior. It alone is not concrete evidence.

Response: Thank you for pointing this out. We have revised the wording for this section. Since the results aren’t concrete evidence, it only suggests the existence of heightened anxiety-like behavior in the two albino strains. (page 5).

  1. It would be beneficial to show the duration of nocifensive behavior over days until resolution rather than just one day.  If testing was done only for 1 day it would be better if that day is consistent between all behavior data. i.e von frey, hot plate and catwalk for day 6 instead of different days.

Response: Both clinical and preclinical studies including ours (PMID: 30807841, 26797506, 28270575) have shown that nocifensive behavior following SCI doesn’t resolve over time. In fact, as we had articulated in the introduction of this paper, neuropathic pain following SCI is a persistent symptom that is hard to treat with conventional means. Moreover, each strain had their own sham mice for comparison of injury effects, so conducting each test for nocifensive behavior on different days is more attainable in practice.

  1. Why was 8 weeks chosen for RNA seq analysis? There was no explanation given, it seems out of the blue.

Response: Mice subjected to contusion spinal cord injury show endogenous recovery during the first few weeks after injury (Fig 2A). When the BMS scores reached to the plateau, a battery of neuropsychiatric behaviors was applied to those animals. After completion of the tests at 8 weeks post-injury, all mice were euthanized at the same timepoint and a portion of the injury tissues were collected for RNAseq analysis. (page 9, page 13)

Reviewer 3 Report

Comments and Suggestions for Authors

Well written and described. Uses well established methods and protocols and looks more in depth at the proposed contributions of genetics in 3 strains of mouse in behavioral recovery after SCI. Makes a relatively but innovative contribution and may as well be useful in promoting further evaluations into the genetics, but equally or more importantly, the microbiome, the transcriptome, proteome, and metabolome differences in strains. Additionally, are there neuroanatomic or connectome differences in the SC between strains? 

Author Response

Well written and described. Uses well established methods and protocols and looks more in depth at the proposed contributions of genetics in 3 strains of mouse in behavioral recovery after SCI. Makes a relatively but innovative contribution and may as well be useful in promoting further evaluations into the genetics, but equally or more importantly, the microbiome, the transcriptome, proteome, and metabolome differences in strains. Additionally, are there neuroanatomic or connectome differences in the SC between strains? 

Response:  We agree with the reviewer that neuroanatomic or connectome analysis of spinal cord tissue will provide more in depth at the contributions of genetics in three strains of mice after SCI. However, we apologize for the lack of relevant data. This project was conducted several years ago. Following the relocation of our lab, many histological and tissue samples for this project were lost in storage. Redoing the model and testing on a new batch of mice would require a strenuous effort, starting with amendments to our current IACUC protocol, ordering new mice from Jackson Laboratories, etc. In total, it would take up to 6 months for additional data.

A literature searches on prior studies show that neuroanatomic and connectome differences do exist between the brains (PMID: 33635426) and sciatic nerve distribution (PMID: 18316160) of different mouse strains. However, the spinal cord seems to be understudied in this area and we hope to delve deeper into this angle in future projects.

Round 2

Reviewer 1 Report

Comments and Suggestions for Authors

I understand why the authors cannot provide histological data.

However, in the new version submitted I don't see the table that should have been added by the authors. 

Author Response

Rev 1: Comments and Suggestions for Authors

I understand why the authors cannot provide histological data.

Response: Much appreciated.

However, in the new version submitted I don't see the table that should have been added by the authors.

Response: Table 1 was uploaded as a separate file in the 1st revision and is now included in page 20 of the revised version.

Reviewer 2 Report

Comments and Suggestions for Authors

Thank you for addressing all of my concerns. I have no further questions. It was a well-written paper to read. I commend your efforts to do pain-staking behavior work on large mice cohorts. 

Author Response

 Rev 2: Comments and Suggestions for Authors

Thank you for addressing all of my concerns. I have no further questions. It was a well-written paper to read. I commend your efforts to do pain-staking behavior work on large mice cohorts.

Response: Thank you!